# Cost-Optimal Planning in the IPC 2018:
# Symbolic Search and Planning Pattern Databases vs. Portfolio Planning

**Stefan Edelkamp** and **Ionut Moraru**
King's College London
{stefan.edelkamp,ionut.moraru}@kcl.ac.uk

## Abstract

The optimal track is one the most exiting events in the International Planning Competition (IPC).

In this position paper we argue that —despite of not winning the competition— symbolic search and pattern databases were likely the most influential planning approaches in the latest IPC in 2018, and, in continuation to the precursor IPC in 2014, should be considered as candidates for the current state-of-the-art.

Five of the Top 6 planners in the 2018 competition, namely Complementary (1 and 2), Planning-PDBs, Symbolic-Bi-directional, and Scorpion are based on these technologies. These planners use the same technology across all domains and plan in one state space.

The winner of IPC 2018 with an $\approx 1\%$ lead in problems being solved, however, is a so-called *portfolio* planner, consisting of a selection of many different planners, one of which is chosen in a classifier that was trained on a manually selected set of benchmark instances. In about half of its successful runs, it called the winner of the previous IPC, which in turn is based on symbolic search. We argue on whether or not to exclude portfolios from the IPC is possible and wanted.

## Introduction

Being inspired by the ultimate goal of general problem solving, in the field of action planning, there is a common understanding that planning competitions ran on a set of unknown and partly new set of benchmark problems, advance planning technology the most.

Starting in 1998, over the years considerable progress has been made in the development of new planners, due to competitors being brave enough to face a coding contest with an unknown benchmark set and with the obligation to offer their source code. Competitors have been confronted with an increasing set of challenges including extended expressiveness of planning domain description language (PDDL), involved planning task metrics, in inherent problem complexity, and in scaling problem sizes. While several tracks were spawned, one is the deterministic part of the IPC, and its track on cost-optimal planning.

The outcome of the optimal track in the most recent 2018 International Planning is revisited in Table 1[1]. The corre-

sponding output on IPC 2014 has been provided and discussed by (Edelkamp, Kissmann, and Torralba 2015).

While most planners present one sole planning technology, the winning planner Delfi (Sievers et al. 2019) is a *portfolio*, a mixture of different technologies. Given a problem task, it selects a planner based on a classifier, being trained on a manually chosen set of known planning benchmark instances. In its set of planners, 16 were contained as part of the Fast Downward planning framework, including at least one (Canonical PDB) that used pattern databases. The most effective approach chosen by this classifier, however, was the symbolic search planning system, which won the precursor 2014 IPC competition. Moreover, in the only domain, where Delfi scored overall clear best, it called this planner.

From the mere planning side, in portfolios there is often little that is novel, the contribution of these planners is often found in the machine learning algorithm, which is trained on a set of known planning problems and which eventually selects the planner configuration to call. In Delfi one planner is called per instance. In terms of the potential of different planning approaches available to Delfi, the IPC outcome with a lead of two more being solved ($\approx 1\%$ of 200 benchmark problems) is quite small, showing that cost-optimal planning is tough, even for portfolios. A change in one domain would have resulted a different outcome.

While portfolio planning was in alignment with the rules of the competition, one underlying issue is some participating planners avoided using portfolio technology, presumably in favour of getting a clearer picture on what technology is currently leading. They seemed to prefer working on new plan search technologies, instead of going in for a mixture of existing planners.

Facing the outcome of the competition and the different type of contributions available in portfolio and non-portfolio planners, people interested in planning especially outside to the core planning community have to be warned not to derive wrong scientific conclusions by only looking at the outcome. Competition results always have to be dealt with care.

This position paper aims to provide a clearer picture on what is the currently leading technology according to IPC 2018 and discusses on whether or not a portfolios help to push or blur the outcome of a competition. The stress of this

---

[1] Readers interested in the planners listed are forwarded to the

according planner abstracts found in the IPC 2018 competition booklet at https://ipc2018-classical.bitbucket.io/planners

| | Agri-cola | Cal dera | Data Net Network | Nuri-kabe | Organic Synthesis | Petri Net Alignment | Sett-lers | Snake | Spider | Termes | Σ |
|---|---|---|---|---|---|---|---|---|---|---|---|
| Delfi1 | 12 | 13 | 13 | 12 | 13 | 20 | 9 | 11 | 11 | 12 | 126 |
| Complementary2 | 6 | 12 | 12 | 12 | 13 | 18 | 9 | 14 | 12 | 16 | 124 |
| Complementary1 | 10 | 11 | 14 | 13 | 13 | 17 | 8 | 11 | 11 | 16 | 124 |
| Planning-PDBs | 6 | 12 | 14 | 11 | 13 | 18 | 8 | 13 | 11 | 16 | 122 |
| SymbBiDir | 15 | 10 | 13 | 11 | 13 | 19 | 8 | 4 | 7 | 18 | 118 |
| Scorpion | 2 | 12 | 14 | 13 | 13 | 0 | 10 | 14 | 17 | 14 | 109 |
| Delfi2 | 11 | 11 | 13 | 11 | 13 | 9 | 8 | 7 | 7 | 15 | 105 |
| FDMS2 | 14 | 12 | 9 | 12 | 13 | 2 | 8 | 11 | 11 | 12 | 104 |
| FDMS1 | 9 | 12 | 10 | 12 | 13 | 2 | 9 | 11 | 11 | 12 | 101 |
| DecStar | 0 | 8 | 14 | 11 | 14 | 8 | 8 | 11 | 13 | 12 | 99 |
| Metis1 | 0 | 13 | 12 | 12 | 14 | 9 | 9 | 7 | 11 | 6 | 93 |
| MSP | 7 | 8 | 13 | 8 | 12 | 10 | 0 | 11 | 6 | 16 | 91 |
| Metis2 | 0 | 15 | 12 | 12 | 14 | 9 | 0 | 7 | 12 | 6 | 87 |
| ExplBlind | 0 | 8 | 7 | 11 | 10 | 7 | 8 | 12 | 11 | 10 | 84 |
| Symple-2 | 1 | 8 | 9 | 7 | 13 | 2 | 0 | 0 | 5 | 13 | 58 |
| Symple-1 | 0 | 8 | 9 | 8 | 13 | 2 | 0 | 0 | 4 | 13 | 57 |
| maplan-2 | 2 | 2 | 9 | 0 | 6 | 0 | 0 | 14 | 1 | 12 | 46 |
| maplan-1 | 0 | 2 | 12 | 0 | 6 | 0 | 0 | 7 | 10 | 6 | 43 |

Table 1: IPC 2018 results, measured in the coverage of benchmark problems, i.e, in the number of tasks solved per domains.

paper, therefore, is to argue that, while not winning the competition because of portfolio planning, the two technologies of BDD-based symbolic search planning (Edelkamp and Helmert 2001) and planning pattern databases (Edelkamp 2001) first joint up by (Edelkamp 2005) seemingly dominated the overall outcome competition.

## Symbolic Search and Pattern Databases

The IPC 2018 planner Symbolic-Bidirectional (SymBiDir) was suggested by the authors and accepted by the organizers as a baseline planning technology. It includes no lower bound at all and, thus, relies on so-called *blind* search, i.e., a search with no heuristic search guidance. As actions carry cost, instead of a breadth-first exploration this induces a cost-first traversal of the state-space graph.

In *symbolic planning*, the core difference to explicit-state space planning is the use of binary decision diagrams (BDDs) to represent state sets in the search compactly (Bryant 1986). As actions can also be represented in form of BDDs encoding the transition relation, it is possible to progress and regress planning state in this succinct functional state set representation to perform forward and backward exploration in an operation called *relational product* (Clarke et al. 1996). A first A\*-type algorithm for BDD-based heuristic search has been proposed by (Edelkamp and Reffel 1998).

One obsevation is that memory savings obtained via the compact representation in a BDD in turn often also lead to significant savings in CPU time. The gain of a symbolic representation in IPC 2018 is amplified, when comparing the performance gap of SymBiDir with the other baseline planner ExplicitBlind. As the names indicate, the two baseline planners are not executing the same exploration, due to the fact that coding regression search is not immediate for the usually given partial goal representation; so that the latter conducts a forward state-space traversal only.

To the contrary, SymBiDir executes bidirectional cost-based search, much in the sense of bidirectional application of Dijkstra's single-source shortest path algorithms (Dijkstra 1959), taking care of the fact that the optimal solution might not be established on the first meeting of both search frontiers. As the BDDs represent state sets, recursive solution construction is needed for extracting optimal plans. Aspects like a partitioned computation of successors (called the *image*), variable ordering based, as well as the inclusion invariant constraints to rule out illegal and dead-end states turn out to be crucial factors to improve the exploration efficiency (Torralba et al. 2017).

The performance results of SymBiDir revealed, that in only two of the ten domains (Snake, Spider) is was not doing well, otherwise the baseline planner would have won the competition! This indicates the power of symbolic state space representation and exploration, and suggests that at least across the entire IPC 2018 benchmark set, the vast amount of refined heuristics for planning do not always lead to a throughout leading technology.

According to the result of the 2018 competition, planning pattern databases (PDBs) (Edelkamp 2001) appears to be one of the few exceptions. On the testbet of IPC 2018 the combination of PDBs and symbolic search in the planners Complementary (1/2) and Planning-PDB are outperforming Symbolic Bidirectional Search. The former two are inspired by results of (Franco et al. 2017), while the latter improves on bin packing algorithms for the pattern selection problem. Besides a major rewrite, one new feature of these new planners is that the forward search is in fact explicit-state, while only the backward traversal is symbolic.

Of course, many heuristics besides PDBs are still worth further investigation. For the case if Snake and Spider were removed, it is difficult to draw a conclusion on different

types of heuristics from the IPC-18 results.

There is no free lunch. But the overall performance of bidirectional symbolic search is surprisingly good, while not using any heuristic. Whether the aspect of bidirectional or the symbolic search contributes the most to this performance, we haven't checked, but we expect one would need both.

SymbBidir performs much better in Agricola than the other planners. Using a PDB heuristics seems to hurt symbolic search and fails in 5 to 9 instances where SymbBiDir succeeds. Some of these problems have been identified and overcome by using *perimeter pattern databases* (Felner and Ofek 2007).

About the bad performance in Spider and Snake there are reasons on why the BDD *explode*, which relate to the subtle ordering problem of dependent BDD variables in grids, This issue has been analyzed and proven to be crucial for representing the goal to ConnectFour as a BDD (Edelkamp and Kissmann 2011), and might be detected fully automatically.

Recall, that planning pattern databases are serving as heuristics and are based on a complete backward exploration in some state-space abstraction. Often a larger number of (hopefully) diverse and complementary patterns are generated and the corresponding databases sought to be combined in an admissible manner to preserve being a lower bound.

While Scorpion is partly a PDB planner performing slightly worse to SymbBiDir in IPC 2018, it showed distinguished performance and scored best in 5 out of 10 domains. Further investigations illustrate that it is performing much better across all IPC benchmarks, i.e., ones including the ones from previous competitions (Seipp and Helmert 2018; 2019). It also has to be added that part of the success of Scorpion is due to Cartesian abstractions combined with a counterexample abstraction refinement (CEGAR) approach (Clarke et al. 2000).

As both heuristics are based on state space abstractions, one may view PDB and Cartesian abstraction heuristics as being related estimates of a similar type. While PDBs and Cartesian abstractions are an important part of Scorpion, much of its strength is in the sophisticated method to *combine* these abstraction heuristics. While the competing BDD based PDB planners mainly use 0/1 cost partitioning a more advanced concept saturated cost-partitioning.

When normalizing the success alongside the domains (some have 150 instances), Scorpion compares well with the top-tier symbolic planning systems: 0.621 (Complementary 2), 0.601 (Scorpion), 0.594 (Planning PDBs), 0.576 (Complementary 1), 0.555 (SymBiDir). Using Delfi in this comparison hardly applies, as for this case the training set overlaps the test set.

Again, all or most planners, even the ones that were not at the top, had some positive results (e.g. being among the planners that solve most instances in some individual domains). Even planners at the bottom have some cases where they perform among the best (e.g. ma-plan in Snake).

## Portfolio Planning

Portfolios erected on existing plan technology, are a recurring pattern in many competitions, and range from restarting strategies, over learning classifiers, to scheduling time slices to existing planners.

Once having fixed the metric and submission instructions, the IPC 2018 organizers felt that they had to follow them. If a planner wins by the metric they decided on before, the competition and the organizers didn't crown it the winner, the teams would rightfully complain.

The coverage metric of problems being solved seemed not to be the issue, so one thing the organizers could have done, would have been to change the rules for submissions. They explained that they had an internal discussion about portfolios early on in the course of running the competition, but decided that the line between a planner with multiple components and a portfolio was too blurry to accurately define. That is why they decided to not have a special rule against portfolios. As one reason given, the LAMA system (Richter and Westphal 2010), one previous IPC winner, might have to be considered a portfolio, because it runs different planners one after the other.

In terms of the organizers of the competition is difficult to set rules that identify portfolios to give them a special treatment, because either they'd be too restrictive and most planners of the competition would be considered portfolios, or they would be too ambiguous, generating complains about what planners are considered to be portfolios. This said, there are certainly many interesting aspects to be learned from portfolios on a per-domain or even per-instance base. Proper portfolio designs with a close-to-optimal choice of planners as in Delfi is a research area on its own.

One way to limit the impact of portfolios in the competition is what Mauro and others have suggested for the Sparkle planning competition[2], where planners are evaluated based on how well they do on individual instances/domains, rather than on getting a good average score. However, this suggestion comes with some issues as well. In particular, the score of a planner completely depends on which other planners are submitted to the competition. Henceforth, if someone submits a version of your planner that works only slightly better, he could get 0 points. One has to wait for the results to see how the approach materializes. Unfortunately, the organizers of this competition are only running the agile track, without insisting on cost-optimal plans. In complexity terms, however, optimality is known to be of crucial importance. For example, finding any plan for many planning benchmarks (such as Blocksworld, Logistics, Sliding-Tile Puzzle) is polynomial. In this case of *satisficing* planning where only plan existence is requested problem tend to be tractable, while the corresponding optimization are often provably hard (Slaney and Thiébaux 2001; Parberry 2015; Helmert 2008).

The emerging set of portfolio and the difficulty of excluding them may be seen as a side effect of the requirement of releasing source code for the planners, as it becomes easier to bundle the planners into one code base. Of course public access to the source code is not a strict necessity for these type of planners.

For some planning researchers, the core issue and concern

---

[2]http://ada.liacs.nl/events/sparkle-planning-19

of portfolio planning is that other researchers use their code, and not so much that the participating planner is a portfolio. So the organizers thought about having had a rule against using code from another research groups. That would have excluded most planners based on Fast Downward planner framework, though, and since they were the majority of the submissions, this would not have been a good idea as well.

The solution Fahiem Bacchus suggested (in personal conversation with the authors) based on his own experience with portfolios in the SAT competitions, is having stated a license that prohibits the use of the code in other tools, would be an option, but then the authors of the planners would have to do so before the competition. In case of planners based on Fast Downward, this, however, is also problematic because such a clause would be hardly compatible with the license of the framework.

While intuitively rather obvious, it is far from simple to distinguish portfolio from non-portfolio planner in a formal definition. One may try to start with the following criteria.

A *planner portfolio* selects, invokes, and possibly terminates different existing planners, based on a trained or hard-coded decision procedure.

This definition may not be a perfect discriminator, as one might be able to transform a portfolio into a non-portfolio without changing the performance by much: just moving the decision procedure further down the line.

It does also not cover a planner that uses the maximum of ten heuristics in an A* search. Some people would like to treat this as a portfolio, because there is no contribution except for the selection procedure of the ten heuristics. At the end, the question remains on when a planner is a novel contribution.

Another suggested definition for identifying portfolio planning is the following.

A *non-portfolio planner* is a single core planning technology, which invokes a plan search in one state space.

But what is with traversing state-space abstractions, which are needed to compute heuristic estimates? Clearly, as highlighted by the IPC organizers, defining portfolios turns out to be intrinsically difficult. There are planners that are clearly portfolios, there are planners that are clearly not, and there is a larger gray area in between.

According to a definition, FF (Hoffmann and Nebel 2001) should not be judged as a portfolio planner. It searches one state space with one heuristic. But FF switches from enforced hill-climbing to best-first search, based on some progress measure. This alone should not classify it as a portfolio approach.

LAMA runs a greedy search based on $\mathrm{h}^{FF}$ and a landmark heuristic (three techniques developed by different authors) and then several weighted A* searches (a different planner and an algorithm also developed by other authors). LAMA may or may not be seen as a portfolio. If it runs three independent searches in parallel, then this may be interpreted as a portfolio technology, but the interconnection of the search is more subtle. LAMA had additional algorithmic contributions on how to move back and forth the states in the different priority queues. If LAMA continues searching the same search space, this is a sign of a non-portfolio. At least it does not start different existing planner.

It is, however, abundantly clear that Delfi is a portfolio planner (not even the authors questions that). It even logs its task-dependent calls to the planner binaries. Delfi actually uses 2 executables, SymBA*, and 16 parameters of selecting planners in the Fast Downward framework. It has a decision procedure trained on a set of manual selected planning tasks. Note that in this setting, we do not count the learning as running, but as programming time.

The performance overhead of portfolio designs can be small. The often criticized effect is that frequently more than 99.9% of the actual running time of a portfolio planner is exclusively spend on existing technology. This is a probably unwanted aspect, which can makes other competitors that contribute non-portfolio planners wondering and reluctant to tune their planners for efficiency. Of course, the size of a contribution must not necessarily be taken in direct correspondence to the profiled time that was spend in running the code added.

By public access to the planners at IPC 2018, one can look at the source code of the contributed planner to validate, on whether or not a planner is in fact a portfolio.

Fast Downward's code base has grown too big, there are pros and cons to that. On the pro side the planner suite is good for benchmarking. For the symbolic search engines in IPC 2018, it was better to use it to combine explicit with symbolic search than sticking to an independent technology in Gamer. In fact, there are myriads of parameters that make Fast Downward behave totally different. Fast Downward is no longer one planner, it is a framework. On the cons side, results on mixing different calls it may blur the messages you to take home.

There were at least two different portfolio planners in IPC 2018: Delfi1 and Delfi2, where Delfi1 was so much better, so that in the following we concentrate on this one, and used Delfi for its shortcut notation. There is published work of the planner authors in the IPC booklet that explain the architecture and the machine learning approach of using deep neural nets in more detail, so that we concentrate on the main aspects. The main idea is to train a classifier on the performance curves of known planning benchmark problems, provided as input images. We had some problems to reproduce the results on our machine, but could look at the competition results. The planners being invoked by Delfi in the IPC 2018 are shown in Table 2. Note that Delfi combines the heuristics listed with symmetry and partial-order reduction.

The story on portfolios will go on. Portfolios have already started integrating the systems from 2018. Essentially, even when pushing the field with new and brilliant ideas, one hardly can win the race against a portfolio, at least in general terms.

While portfolios have dominated some tracks in the IPC (the satisficing track, for example) in the optimal track, the winner of IPC 2014 was not a portfolio. As seen in IPC 2018, there were several planners that got a very close performance to Delfi. Also, some other portfolios participated. Delfi2 and other portfolios (MSP and DecStar could be considered port-

| Approach | Used | Successfully |
|----------|------|--------------|
| SymBA* | 110 | 73 |
| LM cut | 64 | 37 |
| Merge&Shrink | 47 | 20 |
| Canonical PDB | 17 | 13 |
| Blind search | 2 | 2 |
| Total | 240 | 147 |

Table 2: Planners chosen by the Portfolio Delfi1 based on analysing the log files of IPC 2018. Only main planner technologies are mentioned, many more parameters apply to the actual invocation of the code. Note that the number of problems being solved is slightly higher than in the competition outcome, as there were some reformulations of the same problem, where the planner was run, too.

folios as well) were behind many non-portfolio planners. Overall, the results do not show dominance of portfolio planners in general.

What is worse, with winning of the IPC in the pocket, portfolio planners help to acquire project money and to publish in high-ranked journals and conference proceedings, where non-portfolio planners often have a harder time arguing that they are carrying the actual contribution in technology, as especially in research, a second place is often not considered state-of-the-art.

Instead of arguing, whether portfolios should participate or not, we should discuss about how people interpret the results of the IPC and how an analysis that goes beyond *Planner X is the winner* is absolutely necessary. Even planners that are not at the top show that there are domains where they can be really useful. We view this position paper as one step towards this end.

About credits. For a scientific paper it is rather clear that one has to become a co-author, if one contributes substantially to the outcome. With portfolios this is slightly different. In the extreme case, it may happen that the one coder contributes and the other one coder using portfolio technology take the credits for the efficiencies of the work.

Of course we wouldn't ask Hart, Nilsson and Raphael to be co-authors of every forward-search planning paper using A* but we would still cite their work (Hart, Nilsson, and Raphael 1972). Probably the same is true for portfolio planners: they should give credit, where it is due (and the planner abstracts do this).

No question, portfolios also have their own contribution. The contribution in a portfolio planner is the combination of techniques, e.g., how much better is this combination than just running all $n$ components for 1 $n$-th of the time.

In this respect the competition booklet helps a lot as it links the IPC planners to the outside people, but one the other hand, in the scientific race a booklet is no archival publication is rarely counted as a success. This is what one may ask for a portfolio, to be explicit on which planner call achieves which individual performance, and not to bury this information in a lot of other stuff, e.g., on how advanced the machine learning (e.g., deep neural network training) is.

We often insist on a proper publication before the release of the code, but this also does not work for competitions like IPC 2018. The problem is essential, as with the competition the coders provide all source to the public, so we should take more care on who contributes what.

If an outside contribution is dominating the own one consider asking the authors. Sometimes you cite and acknowledge, sometimes you feel this is authorship. For the IPC, we see people taking code, shake it a bit to improve the results slightly, and go on publishing.

Whether or not portfolio planners being trained on sample plans are domain-independent, is also a controversy, especially given that training plan samples selected by hand. Extremist think they are not, but other people may think differently. Surely portfolio planners belong to the learning track. Of course, organizers were quite happy to have that many competitors, and for us it was a tremendous success to see how good our planners performs, even when facing portfolios. We enjoyed to see how hard it is to get some good result in cost-optimal by machine learning.

## Conclusion

This position paper aims at arguing on what is the currently leading technology and discusses whether or not portfolios help to push or blur the outcome of a competition. There is indication but not one definite conclusion alias strong proposal along this paper on how to deal with the given observations, the main purpose of the paper is look behind the scenes and to spawn a discussion. Such discussions have tradition in the IPC. We had a discussion on domain-independence at the emerge of control rules in TL- and TAL-Plan (Bacchus and Kabanza 2000; Kvarnström and Magnusson 2003), we also had a discussion on the effect of hand-coded selection in planners like SG-Plan (Wah and Chen 2004), with complaints on hand-written domain-dependent branching inside the planners' code. Interestingly, the one who complains most about the current IPC organizes the next IPC. Now in 2018 the topic is portfolio planning and the problem of identifying and securing individual planner contributions.

The international planning competition 2018 pushed the field in action planning, set up and executed a well-designed externally controlled experiment, aimed at insights about the true performance of planners, falsified and strengthened hypotheses on essential components, compared different technologies on a common rule set, same architecture, and an agreed input formalism. It awarded scientific prizes and provided opportunities for upcoming publications. The evaluation is much better than what one experiences in conference and journal papers. The results are often surprising, when compared to the wisdom taken from existing publications. Of course, every competition is limited in what it can prove, but its scientific impact is not to be underestimated.

This paper discusses two separate claims: 1) An analysis of the result of the optimal track, claiming that symbolic search and PDBs are leading methods for cost-optimal planning. 2) The advantages and disadvantages of considering portfolios as part of the IPC.

The IPC has always been a competition where the best mix of scientific and engineering skill wins. It has never

been just one good idea that won the competition, but also the skill to implement it efficiently, and even before portfolios there was a chance that the better software developer outperforms the better researcher, possibly even with something that would never be published.

In this case most arguments are made only about IPC 2018 (Delfi vs Complementary/PlanningPDBs). The discussion of what is the place of portfolios in the IPC and other competitions tracks and events is of course a a general one.

What to do? In a competition it is always good to refer to a wider set of planners, but fundamental differences should be highlighted and could have been put into the awarding considerations. It is fine to have portfolios inside the competition, but they should at least be tagged as such, given a portfolio is a different type of planner, and, otherwise, wrong conclusions might be drawn from the event. Otherwise, the competition is doomed to swallow its own core contributors. The risk is that these efforts will die out.

Considering on how many different planning heuristics have been suggested in the past, given that places two to six are symbolic search and planning pattern database planners only, and that the winner of IPC 2018 called a symbolic search planner half of its time is a striking fact. We are not aware of any technology that performs better especially on the 2018 IPC benchmark set. Given fluctuations in the results many planners are playing in the same ballpark.

Recent improvements and simplifications indicate that one can lift the results of symbolic pattern database planning towards winning the IPC 2018 competition post mortem.

While portfolios usually dominate non-optimal IPC tracks, it is indeed a tighter race in the cost-optimal track. Optimal planning seems to be a tough nut to crack for portfolios given the limits in time and space, even when having a bigger toolbox. Portfolios can take on novel contribution for free and quite quickly. The advance one to put on top from event to event is counterbalanced with using many planners at once. Whether such an advance is always possible, is a subject projection. But at least it may be argued that it appears that it might be exhausting for the competitors with non-portfolio planners to come up with novel, original and breakthrough technology at every new IPC and compete with the portfolios of the last one.

**Acknowledgement** Programming is a serious art and comes with a lot of fun. The IPC 2018 is a programming contest that allowed all competitors to impress with stunning and outstanding performance results on yet unseen complex, and diverse problem domains. We thank all competitors for the variety of new planning approaches, advancing the state-of-the-art in many respects. The organizers of the IPC 2018 did a great job, both with the choice/design of the benchmark domains and for running the competition. Eventually, they had to decide on the winner according to the rules set.

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
