# OpenReview forum: "Cost-Optimal Planning in IPC 2018: Symbolic Search and Pattern Databases vs. Portfolio Planning"
_icaps-conference.org/ICAPS/2019/Workshop/WIPC_

### Official Review · AnonReviewer2 · 2019-04-24
**Poorly executed analysis of the state-of-the-art in optimal classical planning following IPC 2018**

**Rating:** 3
**Confidence:** 4

**Review:**

This paper states that it "a) provides a clearer picture on what is the
currently leading technology according to IPC 2018 and b) discusses
whether or not portfolios help to push or blur the outcome of a
competition". I will discuss both of these points separately in the
following.

a) The paper claims that symbolic search and pattern databases are the
dominating technologies in optimal planning following the IPC 2018
results. They reason that 5 of the 6 top performers are built on these
methods. I believe that this is certainly part of the truth, but I
dislike the fact that the authors do not try to analyze the competition
results from a neutral point of view but very obviously pick facts that
make their planners (Complementary and Planning-PDBs) and "their"
techniques (symbolic search and PDBs) look good. Some examples how this
could be done better include:

- The only planner that is analyzed in detail in addition to the
  planners of the authors is Scorpion. However, the explanation of the
  success is (in my opinion) biased: while PDBs are an important part of
  Scorpion, it also uses Cartesian abstractions and, in particular, a
  sophisticated method to *combine* these abstraction heuristics that is
  not mentioned in the paper at all.

- The authors argue that SymBiDir would have outperformed Delfi 1 if it
  weren't for the Snake and Spider domains, and derive that "refined
  heuristics for planning" do "not necessarily lead to a throughout
  leading technology". Then the authors continue by stating that "PDBs
  are one of the few exceptions" because they are "significantly
  outperforming SymbBiDir". However, this is also no longer the case if
  Snake and Spider are removed, so it is not possible to draw a
  conclusion on different types of heuristics from these results.

- It is interesting to analyze results on a per-domain basis, but you
  should not stop when it doesn't favor the message you want to deliver
  anymore. Some interesting further results one could have discussed
  include:
    - without Snake and Spider, SymbBiDir would have won -> are
      heuristics important at all? Is the bidirectional search the
      important part, or is it the symbolic search?
    - without Petri Net Alignment, Scorpion would have won -> explicit
      state space search is competitive with symbolic search, and
      combining (abstraction) heuristics with saturated cost
      partitioning is a promising technique (note that this is also a
      "change in one domain" which "would have resulted in a different
      outcome", but one where "BDD-based symbolic search" performs worse
      than explicit state-space search).
    - without Petri Net Alignment, Data Network and Termes, FDMS and
      Scorpion would have won -> not only Cartesian and PDB abstractions
      are informative, but Merge & Shrink abstractions as well

  Please note that it took me just a few minutes to come up with these
  tidbits. I am certain there are many more interesting facts hidden in
  the results and I am not saying the ones I provide here are the most
  important insights. However, you should analyze the results properly
  if you aim to discuss the state-of-the-art in optimal classical
  planning.

- As a next step, it would have been nice to analyze *why* different
  planners have problems with some domains and are good in others. What
  happened to Scorpion and FDMS in Petri Net Alignment? Why does
  SymBiDir (Scorpion) perform so much better in Agricola (Spider) than
  the other planners? Why does using a PDB heuristic hurt symbolic
  search that much in SymBiDir (failing in 5 to 9 instances where
  SymbBiDir succeeds)? This would have been interesting questions and
  answers that I am missing in this kind of paper, and they are
  certainly relevant to make a statement on the current state-of-the-art
  in optimal classical planning.

- Similarly, it would have been interesting to look at the instances
  that Delfi solves and investigate which techniques lead to its
  success. For instance, how did it achieve the perfect 20 in Petri Net
  alignment, did it switch the algorithm for the right instance or did
  it use something else than Symba* in all 20 instance? Instead of
  trying to learn from Delfi, the authors only argue against its
  inclusion into the competition, but I am certain there are interesting
  things to be learned when Delfi is analyzed on a per-domain or even
  per-instance base.


b) The second part of the paper is on portfolio planners in the IPC.
Even though I have never participated in a classical IPC track, I have
witnessed many discussion on this topic and it seems to be as
controversial as it gets. Even if the community would agree on banning
portfolios, the paper correctly identifies the definition of portfolios
as something that is hard to do. Unfortunately, the paper gives 2
definitions just to show that they would exclude planners from the IPC
that should not be excluded (note that Scorpion is also in the gray area
and might be excluded with the wrong definition), and it fails to
propose a reasonable definition. The same goes for the discussion on
publications, project money acquisition and credits in the papers, where
I am missing a contribution in the form of a proposal how things could
get better.

Even though my opinion is irrelevant, I believe that the IPC has always
been a competition where the best mix of scientific and engineering
skill wins. It has never been just a good idea / good research that wins
the competition, but also the skill to implement it efficiently, and
even before portfolios there was a chance that the better software
developer outperforms the better researcher, possibly even with
something that would never be published.

In my opinion, we should trust that valuable research and well written
papers win best paper awards, and be happy with the fact that it's a mix
of science and engineering that wins the IPC.


Finally, despite thinking that this is a poor paper that should not be
published, I believe that a discussion of the role of portfolio planners
in the IPC could be an interesting one at the workshop. If there is
interest in the topic, I could see a version of this paper accepted
where the authors condense the "middle part" of the section on portfolio
planning into an (approx. 1 page) short paper to enable the discussion.


Further comments:

p1: "The optimal track certainly is the most attracted [...] event in
the IPC" -> I might misunderstand the word attracted, but what is it
supposed to say? It cannot refer to the number of participants, since
there were only 16 participants compared to 22 and 21 in the satisficing
and agile tracks; and if it's supposed to mean that it is "the most
attractive" the word "certainly" is certainly wrong in that sentence
since that is a very subjective thing to say (and I, for instance, would
not share that opinion).

p1: "the teams of other participating planners opted against using
portfolio technology for one sole planning approach to avoid blurring
the scientific outcome in favor of a clearer picture on what technology
is currently leading" -> since I doubt that you talked to all IPC
competitors, it'd be good to make clear here which teams you are talking
about.

p3: There is no "2019 IPC", the Sparkle planning competition does not
use that name.

general: I am no expert on commas in English, but I am certain that
there are too many in this paper. Additionally, going over the paper to
fix (other) grammar errors would help.

---

### Official Review · AnonReviewer3 · 2019-04-25
**The paper brings an interesting discussion but should be refined**

**Rating:** 6
**Confidence:** 3

**Review:**

The paper discusses two separate claims:
  1) An analysis of the result of the optimal track, claiming that symbolic search and PDBs are leading methods for optimal planning.
  2) The disadvantages of considering portfolios as part of the IPC.

This is a topic worth discussing in the workshop, therefore I am in favor of acceptance. But I also think that some things should be changed, down-toning some of the claims:

  - There are statements claiming that the optimal track is the most
    popular/attractive/prestigious of all tracks both in the abstract ("The optimal track
    certainly is the most attracted and probably the most prestigious event in the
    International Planning Competition (IPC).") and introduction ("the probably most
    attractive one is the deterministic part of the IPC, and its track on cost-optimal
    planning."). This is unnecessary and not completely true. Arguably, the track with
    most success is the satisficing track, which have always had more participants than
    the optimal one. Therefore, I think that such statements should simply be removed.

 - Regarding (1), given the results of IPC'18, I'd say that the state of the art methods
    for cost-optimal planning are not only pattern databases, but also other types of
    abstraction and cost-partitioning should be highlighted. In particular:

   * Part of the success of Scorpion is due to Cartesian abstractions with a CEGAR
    approach, and also the saturated cost-partitioning aspect could be highlighted. These
    terms are not even mentioned in the paper and the current description of Scorpion as
    an "an explicit-state PDB planner" is innacurate.
   * Delfi combines the heuristics listed in Table 2 with symmetry and partial-order
    reduction and this should be acknowledged as well.
   * I'd add the references to the IPC planner abstracts of the planners that are discussed.
   * It could also be mentioned that Scorpion performed best in 5 out of 10 domains.
   * I'm not entirely sure what this sentence is trying to say "Perimeter pattern
     databases turned out to be good candidates in this PDB zoo (Felner and Ofek 2007).",
     given that none of the IPC'18 planners uses perimeter PDBs.
   * One could also highlight that all or most planners, even the ones that were not at the top, had some positive results (e.g. being among the planners that solve most instances in some individual domains). Even planners at the bottom have some cases where they perform among the best (e.g. ma-plan in Snake).

  - Regarding (2):

   * The paper should be downtoned a little bit. The discussion of what is the place of
      portfolios in the IPC is a general one, and in this case all arguments are made only
      about Delfi vs Complementary/PlanningPDBs.

   * The paper says "Essentially, even when pushing the field with new and brilliant
       ideas, one hardly can win the race against a portfolio, if there is no ban."  I'm
       not entirely convinced of this. Portfolios have dominated some tracks in the IPC
       (the satisficing track, for example) but in the optimal track, the winner of IPC14
       was not a portfolio. In IPC18, there were several planners that got a very close
       performance to Delfi. Also, some other portfolios participated. Delfi2 and other
       portfolios (MSP and DecStar could be considered portfolios as well) were behind
       many non-portfolio planners. Overall, the results do not show dominance of
       portfolio planners in general.

   * Also, winning the IPC can attract citations, but is far from essential to "acquire
     project money and to publish on high-ranked journals conferences". A second place is
     considered state of the art, and the views that do not acknowledge this should be
     confronted. In my opinion, this is key for this discussion: Instead of arguing
     whether portfolios should participate or not, perhaps we should discuss about how
     people interpret the results of the IPC and how an analysis that goes beyond "Planner
     X is the winner" is absolutely necessary. Again, I'll insist in that even planners that are
     not at the top show that there are domains where they can be really useful.

Minor comments:

 An A*-type algorithms -> algorithm
 often leads -> lead
 These defintion -> This definition
 belated aspect -> related